# Innovative Densification Process of a Fe-Cr-C Powder Metallurgy Steel

**Federico Simone Gobber [1], Jana Bidulská [2], Alessandro Fais [3], Róbert Bidulský [1,4,\*] and Marco Actis Grande [1]**

[1]  Department of Applied Science and Technology (DISAT), Politecnico di Torino, Viale T. Michel 5, 15121 Alessandria, AL, Italy; federico.gobber@polito.it (F.S.G.); marco.actis@polito.it (M.A.G.)

[2]  Department of Plastic Deformation and Simulation Processes, Faculty of Materials, Metallurgy and Recycling, Institute of materials and quality engineering, Technical University of Kosice, Vysokoskolska 4, 04200 Kosice, Slovakia; jana.bidulska@tuke.sk

[3]  Epos S.r.l., Via Pavia 68/72, 10098 Rivoli, TO, Italy; af@eposintering.com

[4]  Asian Innovation Hub, P.O. BOX c6, 04023 Kosice, Slovakia

\*  Correspondence: robert.bidulsky@polito.it

**Abstract:** In this study, the efficacy of an innovative ultra-fast sintering technique called electro-sinter-forging (ESF) was evaluated in the densification of Fe-Cr-C steel. Although ESF proved to be effective in densifying several different metallic materials and composites, it has not yet been applied to powder metallurgy Fe-Cr-C steels. Pre-alloyed Astaloy CrM powders have been ad-mixed with either graphite or graphene and then processed by ESF. By properly tuning the process parameters, final densities higher than 99% were obtained. Mechanical properties such as hardness and transverse rupture strength (TRS) were tested on samples produced by employing different process parameters and then submitted to different post-treatments (machining, heat treatment). A final transverse rupture strength up to 1340 ± 147 MPa was achieved after heat treatment, corresponding to a hardness of 852 ± 41 HV. The experimental characterization highlighted that porosity is the main factor affecting the samples' mechanical resistance, correlating linearly with the transverse rupture strength. Conversely, it is not possible to establish a similar interdependency between hardness and mechanical resistance, since porosity has a higher effect on the final properties.

**Keywords:** powder metallurgy; electro sinter forging; capacitor-discharge sintering; Fe-Cr-C; porosity

## 1. Introduction

The sintering of metal powders is traditionally a purely thermal process where a given amount of compacted powders, called green, is densified in a furnace without applying further pressure. Conversely, in sintering processes such as hot isostatic pressing (HIP), loose powders are densified by the concurrent effect of both high pressure and temperatures. Independently from the process adopted, they are all suitable for obtaining either simple or complex shapes in a near-final geometry [1], thus reducing further mechanical processing, material waste [2], and costs if compared to conventional casting [3] and forging techniques [4]. All thermal sintering processes are characterized by sintering cycles generally lasting 60 min or more. Thermal sintering is commonly used in the manufacturing of gears and automotive components of small dimensions (almost 70% of the total powder metallurgy production) electric contacts but also components for aerospace [5], such as turbine blades [6] and rotors [7].

To reduce sintering times and promote the processing of innovative metallic alloys and metal matrix composites (MMC), the so-called field-assisted sintering techniques (FAST) have been developed in the twentieth century [8]. Pulse plasma sintering has been

successfully employed in sintering Co-based intermetallics [9], while spark plasma sintering has shown the capability of processing metal glasses [10]. Other advanced materials such as Ti-based intermetallics [11], cemented carbides [12], and porous Ti alloys [13] are processable via capacitor-discharge sintering methods. Such technologies are characterized by a reduced processing time (from minutes down to milliseconds for FAST vs. hours for thermal sintering) that allows retaining a very fine microstructure even at the nanoscale [14]. Among the materials commonly adopted in the automotive sector and processed by powder metallurgy, there are no studies on the processing of the 100Cr6 bearing equivalent powder metallurgy (PM) steel. Due to the very low compressibility of this PM steel grade, it would be very challenging to reach high density by conventional press and sintering techniques.

The bulk 100Cr6 bearing steel (equivalent to the AISI 52100 grade) is characterized by high compression and wear resistance, both adhesive and abrasive, with few mechanical deformations occurring even under high cyclic loading. From the heat treatment side, it is quenched in oil, reaching a hardness as high as 64 HRC. Such mechanical properties make use of this material spread in the manufacturing of wear-resistant components such as eccentric gears and cylinders for small cold rolling mills; furthermore, over 90% of ball and cylinder bearings are made from 100Cr6.

Its chemical composition is characterized by high carbon (about 1%) and moderate chromium (1.5%), responsible for the formation of iron–chromium carbides [15]. No rare or costly elements are present in this steel grade; such characteristics make the bulk 100Cr6 the most common and favorable bulk steel due to the balance between cost and mechanical properties.

*Electro-Sinter-Forging (ESF)*

Electro-sinter-forging, or e-forging technique [16], has been demonstrated to be interesting and is gaining importance in the manufacturing of precious alloys metals parts [17], cemented carbides tools [18], memory shape alloys [19], steels [20], and Cu-based alloys [21,22]. The intrinsic advantages of ESF led to the emerging of novel applications and use: one machine is used for forming and sintering to near net shape in a very rapid process requiring less than 10 s per part produced. The amount of energy is limited, and this helps in reducing costs and pollutants. Compared to conventional casting [23] or cutting techniques [24], ESF has low wear of tooling and generally higher precision of parts produced. The technical and manufacturing advantages combine with the possibility of creating innovative materials such as metal composites and diamond composites with high performances, properties, and extremely high densities. The process is relatively simple: a mechanical pulse is superimposed onto an electrical one in a previously loaded die with the powders. A capacitor bank originates the electrical pulse at high voltage; then, a transformer raises the current and lowers the voltage. The electromagnetic discharge is synchronized to the mechanical impulse so that energy is transferred just after reaching a set level of pressure; this guarantees a homogeneous flow of current through the powders. The mechanical pressure compensates for the powder shrinkage during sintering; for this reason, it is raised when the electromagnetic energy is transferred through the powders. The pressure is held from a few milliseconds up to 1–3, then the upper plunger is automatically drawn out of the die, and the lower plunger is moved to the upper part of the die to extract the sintered piece. In this study, a Fe-Cr-C system has been sintered by ESF to prove the efficacy of this peculiar capacitive discharge sintering (CDS) technology in processing a 100Cr6 steel grade that is typically not processed via conventional thermal sintering techniques. The goal of this work is to validate the efficacy of ESF for sintering high-carbon PM steel. This technique was previously used in processing an AISI M2 HSS tool steel starting from pre-alloyed powders [25], but no attempt to sinter steels with higher carbon content is documented in the literature yet. The feasibility of adopting ESF is targeted in this article, establishing a correlation between processing conditions and results from sintering in terms of density and mechanical properties.

## 2. Materials and Methods

Prealloyed Astaloy CrM water atomized powders by Hoganas AB-Sweden have been used as raw material for the study. A detailed analysis of their characteristics is reported in [26]. The chemical composition of the starting powders is given in Table 1.

**Table 1.** Chemical composition of the Astaloy CrM powders [%wt].

| C | Cr | Mo | Fe |
|---|---|---|---|
| <0.01 | 3.00 | 0.50 | Bal. |

Carbon was added either as graphite (powder < 45 μm, commercial product supplied by Sigma Aldrich) or graphene (BET 450 m²/g, commercial product supplied by Sigma Aldrich) to the powders to reach a fraction between 0.95 and 1% wt. A turbula mixer with small amounts of heptane was used to incorporate graphene or graphite with the Astaloy CrM powders. As reported in the powders producer datasheets, the total oxygen content is 0.2%. This indication is important for powders obtained by water atomization, since it can give a fruitful insight into sinterability. ESF is a sintering process activated by electric energy; the higher the powders' oxidation levels, the higher the risk that the current flow is hindered. The Cr content can contribute to creating a higher content of Cr-based carbides that can confer higher wear resistance (especially abrasive wear resistance) to the sintered material. The reason for comparing sintered ad-mixed powders based on Astaloy CrM to bulk 100Cr6 was commercial: Astaloy CrM is specifically used in applications where abrasive wear resistance has to be taken into serious account, thanks to the higher fraction of hard chromium carbides.

For the analysis carried out in this study, rectangular-shaped samples were produced by ESF (20 × 10 × 4 mm). Processing and post-processing parameters are reported in Table 2. During ESF, the specific energy input (SEI) is the parameter closely correlating to temperature, and in a narrow process time, the applied pressure is raised from a starting value ($P_{start}$) to a maximum value ($P_{max}$). Samples submitted to heat treating were austenitized at 850 °C, quenched in oil, and then tempered at 250 °C for 2 h. This heat treatment was performed to compare ESF samples to conventional cast and forged material in the form of cylindrical-shaped samples (10 mm diameter and 4 mm thick) in terms of porosity, microstructure, and mechanical properties.

Electro-sinter-forging was carried out in air without any protective atmosphere. Due to the short processing time (in the order of a few ms), the sintered material does not suffer from surface oxidation or compositional alteration at the surface. As stated in [27], it is impossible to acquire a meaningful temperature profile during the process. A heterogeneous temperature profile related to the current path through the powders is characteristic of ESF. For this reason, it would not be significant to measure temperature just on the core of the sample. Furthermore, the process itself is so fast that a thermocouple would not be meaningfully responsive in such a narrow process window [28].

**Table 2.** Process parameters and post–sintering treatments were employed in this study.

| Sample Series | Powder | SEI [kJ/g] | $P_{start}$ [MPa] | $P_{max}$ [MPa] | Finishing Operations | Heat Treatment |
|---|---|---|---|---|---|---|
| Y | Bulk 100Cr6 commercial | - | - | - | Face milled | Yes |
| B | Astaloy CrM + graphite (0.97% C) | 2.1 | 20 | 220 | - | No |
| C | Astaloy CrM + graphite (0.97% C) | 2.1 | 20 | 220 | Ground (0.05 mm) | No |
| D | Astaloy CrM + graphite (0.97% C) | 2.1 | 20 | 220 | Ground (0.05 mm) | Yes, after grinding |

| | | | | | | |
|---|---|---|---|---|---|---|
| E | Astaloy CrM + graphene (0,97% C) | 1.9 | 20 | 235 | - | Yes |
| F | Astaloy CrM + graphene (0.97% C) | 2.2 | 20 | 281 | - | Yes |
| G | Astaloy CrM + graphene (0.97% C) | 2.2 | 20 | 279 | - | No |

Process parameters were modified through a balance between tool life and material properties. A too high energy input would be detrimental for the ESF press; the risk of melting the core material and breaking the external rim of compacted powders would cause the molten metal to leak out from the core and stick to the plungers, thus reducing their service life dramatically. For these reasons, the strategy adopted in ESF parameters optimization has been to mutually increase the maximum applied pressure ($P_{max}$) and input energy until close-to-theoretical density was obtained. At least three samples for each condition were produced to evaluate TRS and macro hardness, while microhardness and microstructure were evaluated for each series for the sample with the average value of TRS. Samples were formed in a net shape, and only specific samples were lightly ground (0.05 mm) to investigate the border effect of porosity on TRS.

After sintering, the density was derived using the geometrical method. Due to the simple shape of the samples, the volume was measured with a high-precision gauge (Mitutoyo Digimatic Micrometer 293). Some samples were ground by SiC paper to remove 0.05 mm overstock to observe the influence of the surface layer on bending resistance. Mechanical properties were tested with a three-point bending test. Transverse rupture strengths were measured on samples 20 × 10 × 4 mm with a 15 mm distance between the constraints, a 20 N pre-load, and a 0.08 mm/min elongation rate. The specimens were cut in two halves along the yz plane for metallographic preparation (Figure 1).

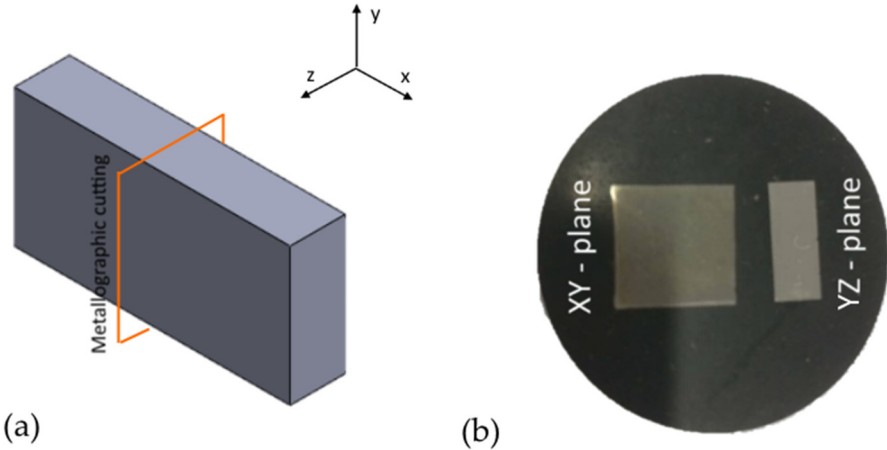

**Figure 1.** Schematic representation of the sample preparation. (**a**) Sample orientation and cutting plane, (**b**) resulting metallographic sample.

The two halves of each specimen were mounted in resin, one half along the xy plane and the other one along the yz plane.

Samples were prepared by grinding with SiC-based abrasive papers (from 200 to 2400 grit) and then polished with cloths soaked with diamond-based suspensions (from 3 down to 1 μm). Both directions, parallel and perpendicular to the loading axis of the ESF machine, were analyzed. Light optical microscopy has been carried out through a Leica MEF4M, and porosity was analyzed by image analysis through the software Qwin (Leica); a total area of 13.19 mm² from five randomly acquired regions of interest was analyzed at 200× magnification per plane (XY or YZ) on each sample. Grain size measurement was

performed just on selected samples, according to ASTM E112. Both micro and macro hardness were performed to investigate the influence of porosity on the microstructure. Vickers microhardness was tested through a Leica VMHT with 500 gf loads, while macro hardness was tested on an EMCOtest M4U 025, adopting the HRN test method. The instrument then converted measured values to the HRC scale. Macro hardness was measured on unmounted samples to prevent the risk of the mounting resin from affecting the measurement. By adopting this micro/macro hardness testing approach, the effect of porosity on microstructure could be separated. Samples in selected conditions were observed by SEM (Zeiss EVO 15 equipped with an Oxford Instruments Ultim Max EDS probe) to characterize their microstructure after ESF and after heat treatment. Carbon and sulphur have been quantified by combustion analysis with the Leco CS-844 analyzer.

## 3. Results and Discussion

### 3.1. Porosity

Porosity was evaluated through image analysis, and the results for the different directions taken into account are reported in Figure 2. Commercial samples of bulk 100Cr6 from casting/forging were not included in the results because their porosity was null. Porosity measured perpendicular to the pressing direction (YZ—plane) is higher than porosity along the pressing direction (XY—plane). This behavior can be attributed to the deformation direction during pressing, contributing to reducing pores along the forging direction.

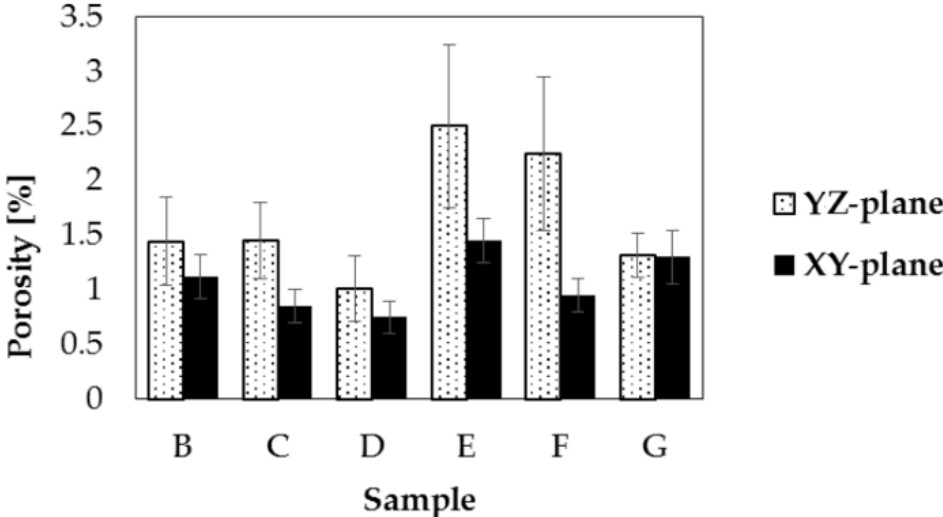

**Figure 2.** Bar chart representing the results of porosity measured on ESF samples.

The evaluation of porosity was carried out after polishing the samples progressively and analyzing porosity over different sample layers to measure a volumic average. Porosity roughly represents the fraction of void volume over total volume. Pore structures like pore size, morphology, and distribution of porosity within the pressed part present critical items in the load-bearing sections, which means the primary controlling mechanism of the mechanical properties [29–32].

A significant aspect to be pointed out is that graphene-containing samples show higher porosity (samples E, F, and G in Figure 2); this evidence is another clue to be considered if a choice between graphite and graphene as the carbonaceous element has to be made.

The micrographs of Figure 3a reveal that towards the edge of the sample, along the Z direction, a relevant degree of porosity of approximately 50–100 μm is detectable both in samples with graphite and graphene (Figure 3b,c).

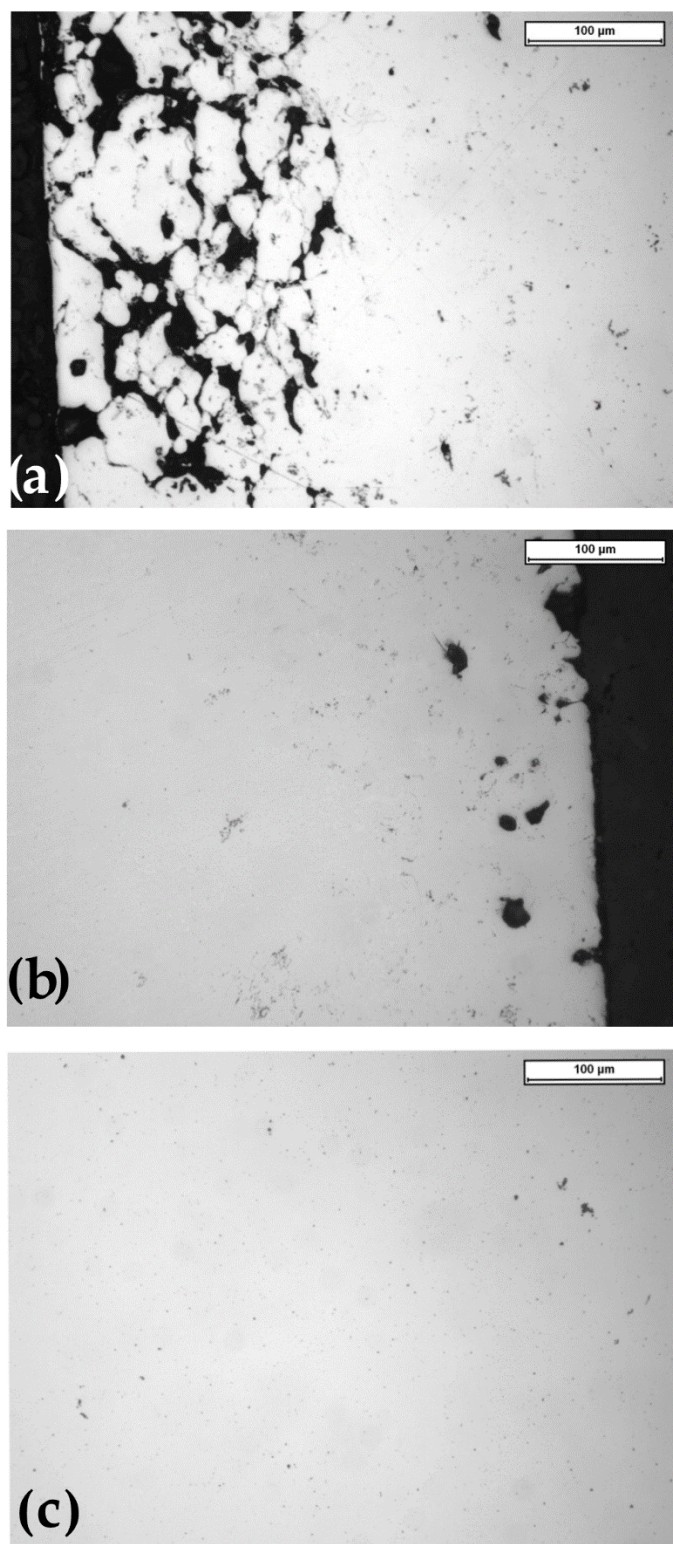

**Figure 3.** Optical micrograph of samples obtained through ESF. (**a**) Edge of sample E (YZ—plane), (**b**) edge of sample B (YZ—plane), and (**c**) core of sample F.

The main difference between the two is that this porous layer shows smaller dimensions for graphite-added powders, significantly affecting the overall degree of porosity in the sample.

Figure 3a explains the high level of measured porosity. Similar considerations can be made for sample F, whose edge is similar to that of E. The core of sample F, on the other hand, is fully dense (Figure 3c), confirming that materials sintered via ESF can suffer from porosity at the edge but not at the core area.

Currently, process parameters need to be further finely tuned: excessive porosity could lead to early failure of the sintered component, although a grinding of 100 μm is considered enough to remove the porous layer.

### 3.2. Microhardness and Macroharndess

Microhardness testing was performed at room temperature, and the results for the tested samples are reported in Figure 4. The distinction is made between planes YZ and XY, Z being the pressing direction and XY the plane perpendicular to it.

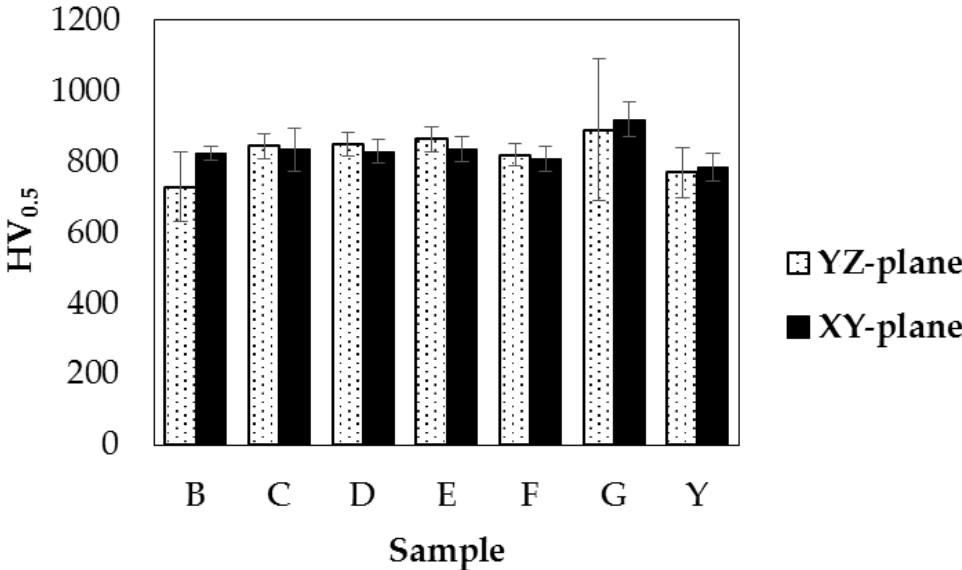

**Figure 4.** Bar chart reporting the microhardness values of the different samples measured perpendicularly and paralleled to the pressing direction.

The measured microhardness values of the ESFed samples are higher on an average value concerning the commercial ones (Y sample in Figure 5) except for sample B. The difference among the samples is confirmed by the ANOVA general linear model ($p = 0.000$; $F = 6.65$) for $\alpha = 90\%$ confidence level. Furthermore, by comparing samples B, C, and D having the same sintering parameters, it is possible to conclude that the heat treatment after ESF does not influence the values of hardness, while grinding increases the average microhardness, probably due to the removal of the outermost layer, where residual pores concentrate more [21].

By comparing samples B and C, differing only for the grinding applied on sample C, a significant increase in the microhardness measured along the pressing direction can be noticed on the ground sample. Samples sintered with the addition of graphene (E, F, and G) show a slightly lower microhardness after heat treatment (E and F). Although sintering conditions were comparable to those applied on samples B to D, where graphite had been added, significant hardness improvements were not observed with graphene. Although microhardness values slightly differ for all samples depending on the direction analyzed, it is just for sample B that this difference is significant from a statistical point of view. A first assessment can be drawn from this analysis: the use of graphene does not seem to produce significant benefits compared to graphite, especially from a cost/benefit perspective. Its cost is 30 times higher than graphite, but the mechanical properties obtained are only slightly superior. Hardness higher than 800 HV in the ESFed state or after ESF + HT is compatible with the hardness of a forged 100Cr6 steel, quenched at 100

°C/s or faster. This observation suggests all the samples have a fully martensitic microstructure already after ESF. 100Cr6 is characterized for the precipitation of $(Fe, Cr)_3C$ [15,16]; these spheroidized carbides precipitate during heat treatment, enhancing the wear-resistant character of the material.

Figure 5 shows the apparent hardness measured in HRC. Samples E and G show significantly lower values, mainly due to pores' presence at the surface. Such porosities cause a drop in the measured hardness while they are not affecting Vickers microhardness. Sample F, heat-treated and containing graphene, reaches the maximum average macro hardness. In contrast, sample E also contains graphene and is heat-treated (but not optimized in the ESF process parameters), which contains a higher porosity, responsible for a lower macro hardness.

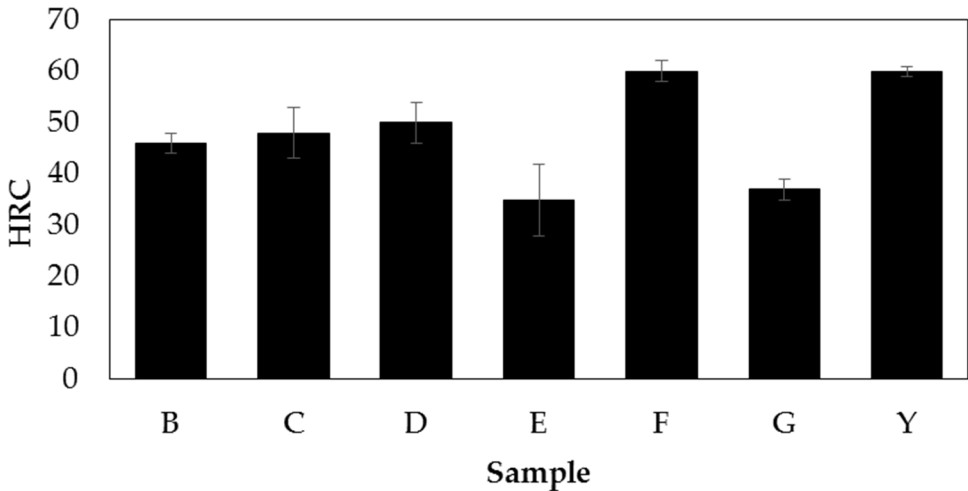

**Figure 5.** Bar chart representing the macroscopic hardness of the samples analyzed (XY plane).

A correlation between TRS and porosity was partially verified by employing statistical tools (Figure 6). A linear regression model was applied at a confidence level $\alpha = 90\%$ to TRS values measured experimentally. From the statistical evaluation, it can be concluded that a linear relation exists between TRS and the levels of porosity on the YZ plane ($p\_$value = 0.012). This correlation is expressed by the linear mathematical relation (TRS = 1715–567.9 * %porosity) and has predictive efficacy for explaining 74.55% of the experimental data variation. Despite having reported the data in Figure 6, it was impossible to calculate a statistically significant linear relation between TRS and %porosity on the XY plane. At a confidence level $\alpha = 90\%$, the significance level is just $p = 0.108$. This misfit can be accounted for by outliers, but the authors did not want to discard any experimental evaluation, since, at this preliminary stage of research, it is considered important to have a broad scenario on all the possible failures, both explainable and non-directly explainable. Such observations are useful for further characterizing a specific and narrow window of process parameters.

From this evaluation, according to the literature, it seems that the porosity in the direction perpendicular to the plungers (YZ-plane) has a relevant effect on TRS. From this observation, it can be said that an increasing level of porosity negatively affects mechanical properties. From a provisional perspective, it is possible to predetermine the TRS of an ESFed Astaloy CrM sample based on a linear relation with a certain accuracy.

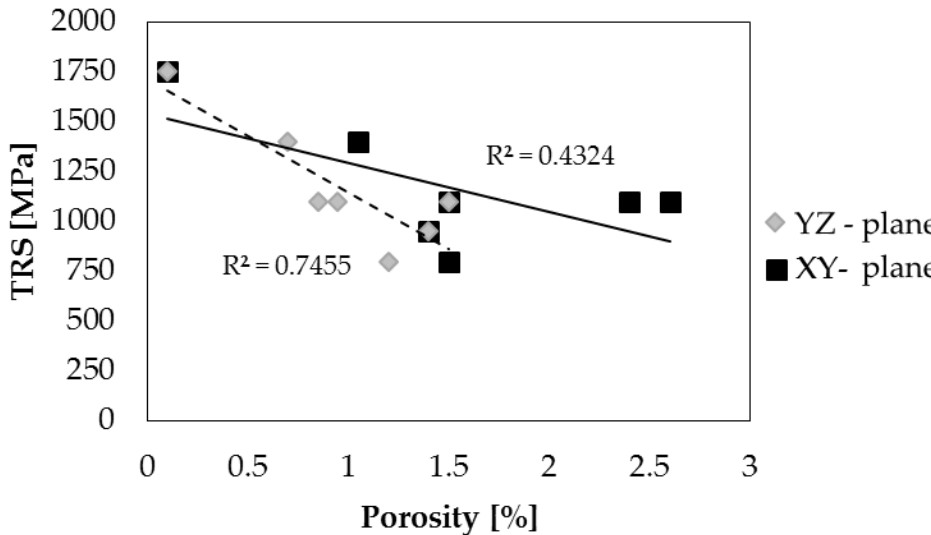

**Figure 6.** Correlation between porosity in the direction of pressing and transverse rupture strength.

### 3.3. Microstructure and C and S Content

Chemical etching with Nital 2 evidenced the microstructure of the samples. Samples B and G, whose microstructures are shown in Figure 7, were molten at the core but not at the surface. If the whole compact of powders had melted during ESF, the plungers' high impulsive forces would have squeezed the liquid out of the die, thus damaging the tooling. The choice of the process parameters is a crucial part of ESF to prevent undesirable failures and compromising the dies. Too low values of the process parameters are not helpful in densifying the powders' compact, while too high parameters (especially SEI) can melt the whole compact of powders with the risk of welding the material under process and the die.

The surface layer (approx. 150 μm) is mainly ferritic (light-colored), having a certain degree of retained pores (Figure 7a). This observation is not intuitive if we consider the composition of the Astaloy CrM + graphite. The microstructure of steel with 1% carbon would mainly be martensitic or pearlitic; no isolate ferrite should be present. In this case, the temperature at the edge of the ESFed sample does not reach the limit for diffusion and alloying to occur. The temperature reached at the interface between plungers and powders is lower than that reached at the core of the sample, so the mix of Astaloy CrM and graphite does not alloy completely, with graphite and metallic powders staying separate and leading to a higher degree of surface porosity. As for other ESFed materials [27], a core–rim structure is typical and is not related to oxidative processes.

Moving towards the core of the sample (Figure 7b), a dendritic-like microstructure is observable, with dendrites developing toward the direction where heat is dissipated. This microstructure is caused by the high amount of heat that locally melts the internal part of the loose powders contained in the die and dissipates towards the plungers and the die itself. The core is melted instantaneously and rapidly solidified again. The occurrence of melting in ESFed materials was also observed in previous studies on NiTi alloys [27]. The microstructure observed in Figure 7c has some trace of directional solidification, as do microstructures deriving from casting, despite being significantly much finer compared to a conventional microstructure from casting.

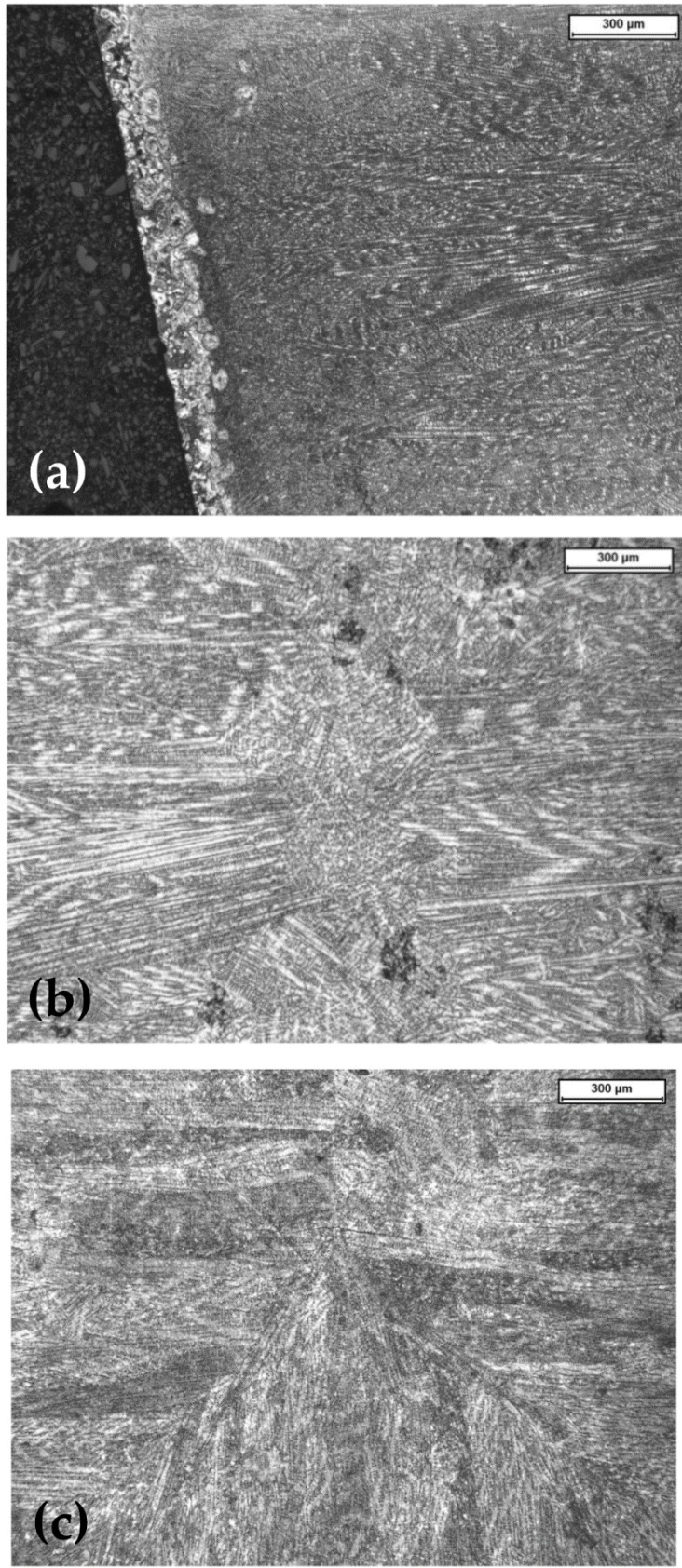

**Figure 7.** Optical micrograph of the microstructure of samples obtained through ESF on the YZ plane. (**a**) edge of sample B, (**b**) core of sample B, (**c**) core of sample G.

The heat treatment's primary purpose was to observe if it is possible to increase the material's mechanical properties through a conventional quench and temper. After heat treatment, a relevant increase in TRS is observable (through comparison between sample C and sample D). These two samples have in common the same processing parameters and post-processing treatment (0.05 mm face milling to remove the outermost porous rim) except for the heat treatment that was performed on sample D only.

After heat treatment, a relevant increase in TRS is observed (1080 ± 162 to 1340 ± 147 MPa), but a further increase of 34% in TRS is necessary to obtain results comparable to the average of the cast and forged materials (C&F). Conversely, if we consider the high standard deviation for the TRS of the C&F material (1790 ± 499 MPa), it can be inferred that mechanical properties close to those even at this preliminary stage of the C&F have been reached in the ESFed + heat-treated condition. Another relevant feature that needs to be highlighted for the as-ESFed samples is the process's too-high hardness. A hardness level above 800 HV, proper to high carbon steel after quenching, is reached after electro-sinter-forging the admixed powders. This experimental observation matches with the temperature-related hypothesis: (1) the maximum thermal flow passes through the core of the sample, and there the temperature rises instantaneously; (2) the fully densified sample is then cooled down very quickly due to the small mass of the sample and the tight contact with the copper plungers; (3) the sample is extracted from the ESF press bare-handed, at a temperature allowing it to be handled safely.

A peak temperature above 850 °C should be reached at the core of the material. Above that temperature, the material is fully austenitic, and a suitable rapid cooling would cause the martensitic transformation to occur, thus increasing the hardness of the material.

Another relevant contribution that has to be taken into serious account for the densification mechanism is that electric current, except the heating effect, has the characteristic of promoting the electroplasticity effect [33].

Significant differences arise from comparing the conventional forged microstructure to a 100Cr6 steel (Figure 8a) and ESFed steel (Figure 8b–d). After quenching and tempering the forged 100Cr6, its microstructure reveals a martensitic matrix in which carbides are dispersed, while the presence of an almost cellular, fine dendritic morphology characterizes the as-ESFed microstructure. Similar to what is documented for other high carbon steels processed by rapid solidification techniques [34], the microstructure of the ESFed steel (Figure 8b,c) consists of a dark-colored martensitic matrix surrounded by a light-colored network [34]. Based on literature references for rapidly solidified high carbon steels such as AISI M2, having a comparable quantity of carbon, the microstructure in the as-ESFed state is martensitic in the dark-colored matrix and austenitic in the light-colored network [34]. The formation of martensite plates and retained austenite is confirmed in [35] for both an unalloyed high-carbon C90 steel and an AISI M2 steel when rapidly solidified. These two steel grades have a carbon content similar to 100Cr6 steel but differ radically in the alloying elements' content. The difference in alloying elements' content affects just the relative fraction of martensite and austenite after rapid solidification. The matrix grain size is 8.56 ± 2.18 μm, and it was measured from the micrographs. The formation of this microstructure is related to the rapid solidification occurring during electro-sinter-forging; indeed, the liquid phase formation during ESF is finally documented due to the presence of the fine dendritic morphologies observed. Dendrites are not formed other than during solidification.

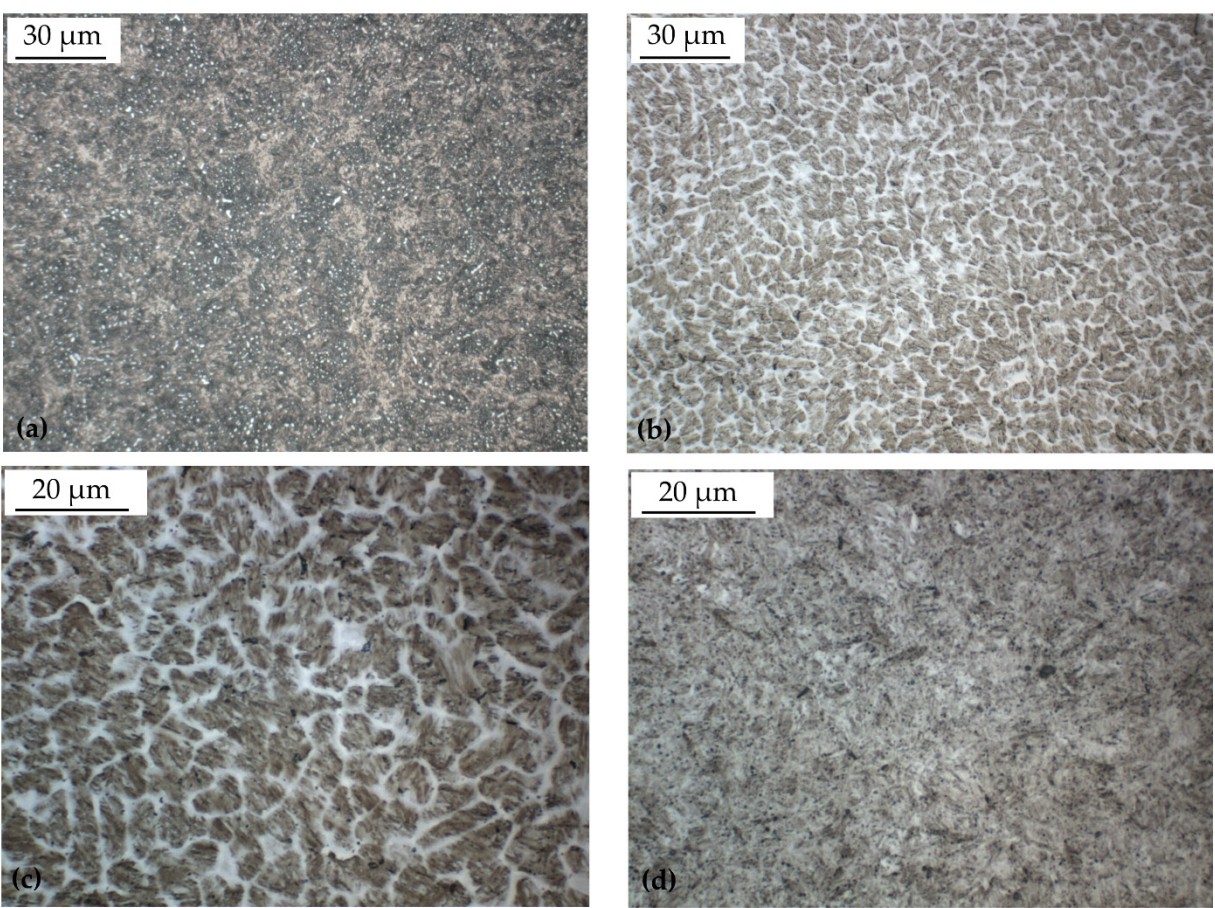

**Figure 8.** Optical micrographs of the microstructure of samples on the YZ plane. (**a**) 100Cr6 steel quenched and tempered (Q + T), (**b**,**c**) sample C as-ESFed, and (**d**) sample D in the Q + T condition.

This microstructure confers high hardness to the as-processed material due to the contemporary reinforcement effect of martensite and the grains' fine size.

By SEM imaging, some micro porosities are observed (Figure 9a); after densifying the powders by ESF, the morphology of the martensite is fine despite the high carbon content and can be referred to as fine lath martensite (Figure 9c). The network morphology of the fine dendritic microstructure is kept even after quenching and tempering the steel, as observed in Figure 9b.

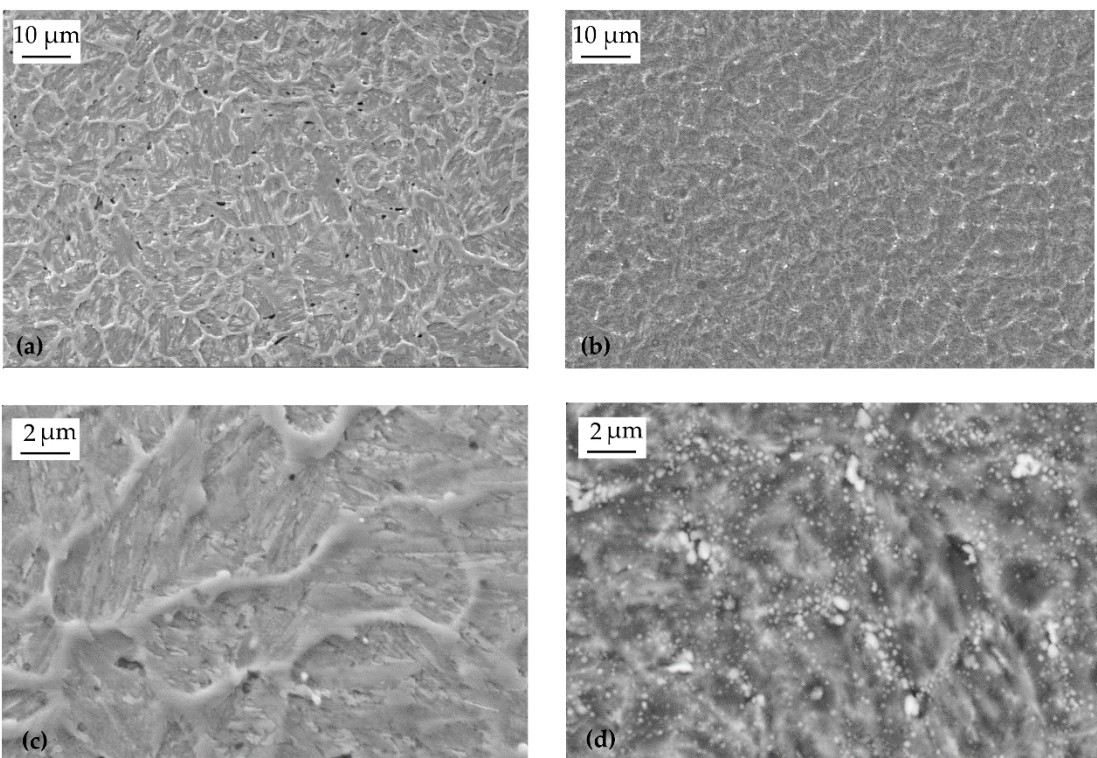

**Figure 9.** Scanning electron microscope (SEM) micrographs of the microstructure of samples obtained through ESF on the YZ plane. (**a**,**c**) Sample C in the as-ESFed condition and (**b**,**d**) sample D in the Q + T condition.

According to martensite tempering mechanisms, a diffused dispersion of tempering carbides precipitates during heat treatment (Figure 9d), while carbon is diffused out of the martensite. The precipitation of micrometric carbides mainly occurs in the former network (Figure 9b), while sub-micrometric carbides diffusely precipitate inside the former martensitic matrix (Figure 9d). The compositional assessment through EDS (Figure 10) confirms that carbides are FeCr-based. The fine size of the analyzed carbides makes the measurement less precise, affected by the surrounding matrix. Nevertheless, the comparison between Cr at% in the three spots and the area measurement confirms that the bright particles are FeCr-based carbides. The fine microstructure of sample D (Figure 8d, Figure 9d) justifies its superior hardness (Table 3) when compared to the forged material (Figure 8a). On the contrary, small porosities act as a stress concentrator, thus reducing the TRS of the ESFed material.

The process of admixing Astaloy CrM powders with either graphite or graphene results in a carbon content equal to 0.94 %wt, compliant with the EN ISO 683-17 standard (0.93 ÷ 1.05 %wt), and a low sulphur content (<0.00047%).

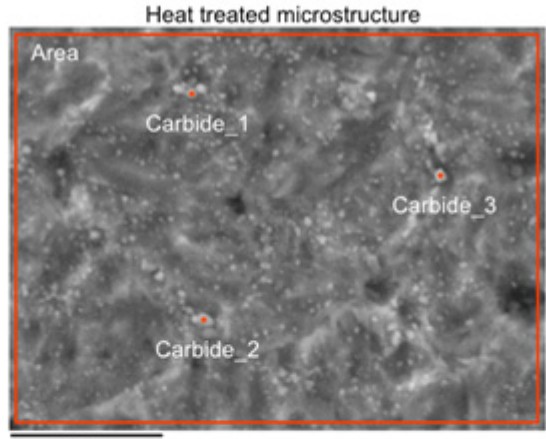

| ROI | C | Cr | Mn | Fe | Mo | Total |
|---|---|---|---|---|---|---|
| Carbide_1 | 16.31 | 3.80 | | 79.78 | 0.10 | 100.00 |
| Carbide_2 | 17.61 | 4.46 | 0,37 | 77,32 | 0.23 | 100.00 |
| Carbide_3 | 20.23 | 5.15 | | 74.49 | 0.13 | 100.00 |
| Area | 17.16 | 1.82 | | 80.92 | 0.10 | 100.00 |

**Figure 10.** EDS analysis on sample D in the Q + T condition.

**Table 3.** Synoptic table of the mechanical properties of the ESFed samples.

| Sample | Density [g/cc] | TRS [MPa] | Hv 0.5 kg | Post-Processing |
|---|---|---|---|---|
| Y | 7.8 | 1790 ± 499 | 810 ± 62 | Face milled + HT |
| B | 7.79 | 798 ± 146 | 863 ± 45 | - |
| C | 7.79 | 1080 ± 162 | 850 ± 56 | Grinded (0.05 mm) |
| D | 7.79 | 1340 ± 147 | 852 ± 41 | Grinded (0.05 mm) + HT |
| E | 7.89 | 951 ± 86 | 863 ± 41 | HT |
| F | 7.87 | 658 ± 112 | 914 ± 155 | HT |
| G | 7.87 | 1050 ± 105 | 823 ± 39 | - |

Although several works are present in the literature regarding compositions similar to those tested in this research, the main results of the most significant studies were reported in Table 4. All the articles found in the literature analyze the properties of materials sintered with press and sinter. No references were found about electric field-assisted techniques used to sinter Astaloy CrM or similar powders. Results comparable to those presented in this study were obtained in [36] through a conventional press and sinter process lasting 30 min at the sintering T of 1240 °C, in a reducing atmosphere ($90N_2/10H_2$); samples were then sinter-hardened and tempered. Considering that the present work results and those from [36] are similar in terms of mechanical properties, it is paramount to evaluate the processing time, energy consumption, and cost-related issues. It is clear that, currently, ESF has an evident limit in the maximum size of the component that can be densified [18]. However, ESF shows its large potentialities in short processing times and energy consumption concerning press and sinter.

**Table 4.** Literature data of powder metallurgical materials deriving from admixed Astaloy CrM powders, compacted through press and sinter techniques.

| Ref. | Composition | Density [%] | Hardness [HV] | Strength [MPa] |
|---|---|---|---|---|
| [36] | Astaloy CrM + 0.8 %wt graphite | 7.2% | 230 | 325 (UTS) |
| [37] | Astaloy CrM + 0.6 %wt graphite | 7.81% | 432 | 1470 (UTS) |
| [38] | Astaloy CrM + 0.6 %wt graphite | 7.2% | 280 | 2400 (compressive) |
| Present work | Astaloy CrM + 0.97 %wt graphite | 7.79% | 852 | 1340 ± 147 (TRS) |

## 4. Conclusions

In this experimental work, ESF was used to densify a PM steel grade with a composition close to AISI 52100, commonly known as 100Cr6, a material conventionally obtained by casting/forging only.

The following points represent the main achievements obtained:

- Astaloy CrM powders were successfully mixed and then alloyed with graphite or graphene to obtain samples with the same carbon content of 100Cr6 steel. Both graphite and graphene effectively raise the carbon content in the starting powders, but based on the compromise between the cost and performance of the materials, it is reasonable to suppose that graphite can provide a more proper and affordable solution.

- ESF's typical microstructures were observed in the processed samples, presenting a core–rim microstructure distinctive to ESFed materials. By properly tuning the process parameters, fully dense material is obtainable. A careful evaluation of parameters is needed to densify the material without damaging the machine. Surface finishing the sintered samples by grinding must be taken into account to remove the fraction of porosity concentrated in the material's outermost layers.

- High values of hardness compatible with a quenched material were observed after ESF. Heat treating was not effective to further increase the hardness of the investigated material.

- A linear correlation between porosity on the YZ plane and TRS was found for the tested samples.

- Although further characterizations are undergoing, the results from this study show that discharge sintered 100Cr6 from an ESF process can be comparable to a forged product. The amount of energy and the time needed to produce the solid material with the proposed method is a clear environmental and industrial advantage.

**Author Contributions:** writing—original draft preparation: F.S.G. and A.F.; writing—review and editing: M.A.G. and J.B.; investigation: F.S.G., R.B., and J.B.; supervision: A.F. and M.A.G.; methodology: M.A.G. and A.F.; validation: J.B. and M.A.G.; formal analysis: F.S.G.; resources: A.F. and M.A.G.; funding acquisition: J.B. All authors have read and agreed to the published version of the manuscript.

**Funding:** This research was partially co-funded by VEGA, grant number 1/0599/18.

**Acknowledgments:** The authors would like to thank Luca Capurso for his precious work in the execution of the analysis.

**Conflicts of Interest:** The authors declare no conflict of interest.

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
