# Peer review of "Innovative Densification Process of a Fe-Cr-C Powder Metallurgy Steel"

_metals, doi:10.3390/met11040665_

Round 1

Reviewer 1 Report

Average paper, numerous modifications are required.

  1. The size of graphite and graphene particles is not given.
  2. The sample orientation is not clear. The figure showing the relation between sample edges and axis would be very helpful. Page 5, line 171: pressing direction is Z. Page 8, line 260: porosity in the direction perpendicular to the plungers (XZ). Both statements are contradictory.
  3. Table 2: SEI, Pstart and Pmax are not explained. Does it mean that the pressure increases from 20 to 220 MPa within a few milliseconds?
  4. The phase identification is based only on literature data (Fig.7a – surface layer of ferrite, Fig. 7b – dendritic-like structure. But what about phase composition? There are dark and light areas.
  5. The same remark on Fig. 8. There is martensite and carbides after heat treatment performed, however this is a suppose only. Unfortunately not one paper is cited in this paragraph for confirmation of authors’ speculations. The description that dark regions correspond to martensite and light ones – to austenite is not sufficient. The simplest XRD studies are necessary.
  6. The method/software used for grain calculations is not given.
  7. The method of density measurement (Table 3) is not given.
  8. The conclusions related to the influence of graphite/graphen addition and processing parameters on microhardness are too strong. Practically the HV0,5 values obtained for all samples are inside the experimental error.

Author Response

Answer to Rev 1

At first the authors would like to thank the reviewer for the valuable comments which helped in the improvement of the quality of the paper

Q1. The size of graphite and graphene particles is not given.

A1. All the information regarding graphite and graphene (commercial products) were added to the article.

Q2. The sample orientation is not clear. The figure showing the relation between sample edges and axis would be very helpful. Page 5, line 171: pressing direction is Z. Page 8, line 260: porosity in the direction perpendicular to the plungers (XZ). Both statements are contradictory.

A2. A figure detailing the relation between sample edges pressing axis was added (Figure 1). The contradictions have been clarified and all the series in the different Figures have been consolidated by using the sole notations “XY - plane” and “YZ – plane”.

Q3. Table 2: SEI, Pstart and Pmax are not explained. Does it mean that the pressure increases from 20 to 220 MPa within a few milliseconds?

A3. As correctly mentioned, this is exactly what happens during the electro sinter forging. Thanks to your comment, the meaning of each parameter was detailed inside the article (materials and methods).

Q4. The phase identification is based only on literature data (Fig.7a – the surface layer of ferrite, Fig. 7b – dendritic-like structure. But what about phase composition? There are dark and light areas.

The same remark on Fig. 8. There is martensite and carbides after heat treatment performed, however, this is a supposed only. Unfortunately, not one paper is cited in this paragraph for confirmation of authors’ speculations. The description that dark regions correspond to martensite and light ones – to austenite is not sufficient. The simplest XRD studies are necessary.

A4. EDS analyses were performed on the heat-treated material. The results are presented in Figure 10. From the high values of hardness measured by microhardness, it is not possible to refute that martensite is present both in the as-ESFed and after heat treatment. This hypothesis is confirmed by the microstructural analysis (Figure 8 and Figure 9). FeCr-based fine carbides precipitating after heat treatment were analysed by EDS and some further literature reference was added as a support. Based on literature references for rapidly solidified high carbon steels such as AISI M2, having a comparable quantity of carbon, the microstructure in the as-ESFed state is martensitic in the dark-colored matrix and austenitic in the light-colored network. The formation of martensite plates and retained austenite is confirmed in literature for both an un-alloyed high-carbon C90 steel and an AISI M2 steel when rapidly solidified. These two steel grades have a carbon content similar to 100Cr6 steel but differ radically in the alloying elements content. The difference in alloying elements' content affects just the relative fraction of martensite and austenite after rapid solidification.

Q5. The method/software used for grain calculations is not given.

A5. The information about the method adopted for performing porosity and grain size measurements were added to the Materials and Methods section.

Q6. The method of density measurement (Table 3) is not given.

A6. The information about the method adopted for performing density measurements was added to the Materials and Methods section.

Q7. The conclusions related to the influence of graphite/graphene addition and processing parameters on microhardness are too strong. Practically the HV0,5 values obtained for all samples are inside the experimental error.

A7. Assuming that the Reviewer refers to the sentence “A first assessment can be drawn from this analysis: the use of graphene does not seem to produce significant benefits, especially from a cost/benefit perspective”, the authors would clarify that the comparison is relative and not absolute. With this sentence, the authors meant to say that there is no evident advantage in using graphene compared to graphite, from the point of view of mechanical properties. Then, due to the high cost of graphene, it is even less desirable its adoption. According to your comment, this concept was made explicit in the article.

Reviewer 2 Report

This work addressed the mechanical properties and porosity of powder metallurgy Fe-Cr-C steels. In general, it is a good work and well-written. Here, I have some suggestions on the current version.

1) You  had better to show how to prepare your metallographic samples and where is the position of interest to observe.

2) how many fields or cross-sections do you measure when you calculated the percent of porosity?

3) Although you have said the porosity has a signification effect on mechanical properties, we do not see how they affect the fracture or hardness. So, you must show the fractographs and other support evidence.

Author Response

Answer to Rev 2

Dear Reviewer,

Thank you for your thoughtful comments and for the time spent reviewing our article. Below the answers from the authors are reported.

Q1. You had better show how to prepare your metallographic samples and where is the position of interest to observe.

A1. A figure detailing the relation between sample edges pressing axis was added (Figure 1). The contradictions have been clarified and all the series in the different Figures have been consolidated by using the sole notations “XY - plane” and “YZ – plane”.

Q2. how many fields or cross-sections do you measure when you calculated the per cent of porosity?

A2. The information about the method adopted for performing porosity and grain size measurements were added to the Materials and Methods section.

Q3. Although you have said the porosity has a significant effect on mechanical properties, we do not see how they affect the fracture or hardness. So, you must show the fractographs and other support evidence.

A3. In this article's frame, there is the feasibility study of an innovative processing alternative for a near-net-shape production method for 100Cr6 steel. Further studies devoted to fracture mechanisms' characterization are undergoing and will be part of a focused article. The authors modified the sentence, and the effect of porosity on mechanical properties is proposed as a hypothesis.

Reviewer 3 Report

Please, see the file attached.

Author Response

Answer to Rev 3

Many thanks for your valuable comments aimed at the improvement of the paper.

Please find the remarks to your points as follows:

Q1. I will not separate metallurgy in the title, the hyphen makes complex the idea.

A1. Unfortunately it is a formatting issue, we will try to ask MDPI if perhaps it is possible to reduce the font size to have the word “metallurgy” on the same line.

Q2. Metallurgy production [2–4, 7–9], must be forbidden. Please go carefully in each work for extracting the real ideas. In some academies like Chinese ones, this style is not well welcome.

A2. That kind of quoting was modified according to the reviewer’s suggestion.

Q3. Figure 1 is a picture…just a picture…it does not show anything interesting. 

A3. The image was removed because it was suggested it was not useful in clarifying the sintering process. A more suitable figure was added describing the preparation method adopted to mount the samples

Q4. Astaloy table is yours or from standards? Did you perform a proper chemical quantification.

A4. Astaloy table comes from standards supplied by the producer. A quantitative chemical analysis was performed on sintered samples to ascertain the nature of carbides and was added to the final part of the article.

Q5. There is not any reference from China or Asia works…no ant MDPI paper…why not?

A5. According to the reviewer’s suggestion, some works from Asian academies were added to the text of the paper

Q6. Figure 6. Correlation between porosity in the direction of pressing and transverse rupture strength. How many testpiece at each force level…statistics… Border effects could affect the results as it was tested in the society Society of Experimental Mechanics and publishes by Krahmer in the journals Experimental Techniques. Experimental Techniques 40 (6), 1555-1565 even if they did not include porosity test pieces as coupons. Please discuss the way of making finishing and extratecting the testpieces.

A6. Three samples for each condition were produced to evaluate TRS but then hardness and microstructure were evaluated for the sample with the average value of TRS. Samples were formed net – shape and, only specific samples were lightly ground (0.05 mm) to investigate the border effect of porosity on TRS. This sentence was added to the materials and methods chapter.

Q7. Your technology is very interesting. Why did you mention Graphene in the abstract.

A7. Graphene was mentioned in the abstract because it was used as an alloying element to increase the Carbon fraction in Astaloy powders to get a composition close to 100Cr6. Both graphite or graphene were used, and finally, due to a non-significant difference between the addition of graphite or graphene. As a conclusion, graphite was chosen due to its lower cost.

Round 2

Reviewer 1 Report

The Authors have improved the quality of their paper introducing recommended changes.

Adding a figure showing the sample orientation and detailed description of phase composition based on EDS results was a beneficial step.